# Kiwifruit Exchanges for Increased Nutrient Richness with Little Effect on Carbohydrate Intake, Glycaemic Impact, or Insulin Response

**DOI:** 10.3390/nu10111710

**Published:** 2018-11-08

**Authors:** John Monro, Kerry Bentley-Hewitt, Suman Mishra

**Affiliations:** 1New Zealand Institute for Plant & Food Research, Private Bag 11600, Palmerston North 4442, New Zealand; kerry.bentley-hewitt@plantandfood.co.nz (K.B.-H.); suman.mishra@plantandfood.co.nz (S.M.); 2Riddet Institute, Massey University, Palmerston North 4442, New Zealand

**Keywords:** kiwifruit, carbohydrate exchanges, glycaemic response, glycaemic glucose equivalents, vitamin C

## Abstract

Background: Kiwifruit are nutrient-rich and have properties which indicate a low glycaemic impact compared with many cooked cereal foods, suggesting that they may be used for dietary enrichment of vitamin C without glycaemic cost. Aim: To develop tables for equi-carbohydrate and equi-glycaemic partial exchange of kiwifruit for glycaemic carbohydrate foods. Method: The available carbohydrate content of Zespri^®^ Green and Zespri^®^ SunGold kiwifruit was determined as sugars released during in vitro digestive analysis. Glycaemic potency was determined as grams of glucose equivalents (GGEs) in a clinical trial using 200 g (a two-kiwifruit edible portion) of each cultivar, non-diabetic subjects (*n* = 20), and a glucose reference. GGE values were also estimated for a range of carbohydrate foods in the New Zealand Food Composition Database for which available carbohydrate and glycaemic index values were available. The values allowed exchange tables to be constructed for either equi-carbohydrate or equi-glycaemic partial exchange of kiwifruit for the foods. Results: GGE values of both kiwifruit cultivars were low (“Hayward”, 6.6 glucose equivalents/100 g; “Zesy002”, 6.7 glucose equivalents/100 g). Partial equi-carbohydrate substitution of foods in most carbohydrate food categories substantially increased vitamin C with little change in glycaemic impact, while equi-glycaemic partial substitution by kiwifruit could be achieved with little change in carbohydrate intake. Conclusion: Equi-carbohydrate partial exchange of kiwifruit for starchy staple foods is a means of greatly increasing nutrient richness in a diet without the physiological costs of increased glycaemia and insulin responses or carbohydrate intake.

## 1. Introduction

Kiwifruit is one of the most nutrient-rich of readily available fruits, and can make a valuable contribution to dietary intakes of micronutrients and phytochemicals that foster good health through a variety of protective mechanisms. So much so that it has recently been recommended that kiwifruit should be considered as part of a natural and effective dietary strategy to address some of the major global health and wellness concerns [1]. One kiwifruit per day has been shown to be sufficient to achieve “healthy” plasma levels of vitamin C [2] and to saturate muscle tissue vitamin C concentrations [3].

Fruits are also generally rich in available carbohydrate in the form of approximately equal proportions of glucose and fructose. In kiwifruit, for instance, sugars make up about 12% of the edible portion of the fresh fruit, consisting of glucose, fructose, and sucrose (about 2:2:1), and represent a major proportion of the skin-free dry matter (New Zealand Food Composition Database, 2015). Consequently, with an increasing global incidence of diabetes and obesity, the high content of sugars is often seen as reason to avoid fruit, ignoring the fact that fruit phytochemicals, including vitamin C, may have a valuable role in counteracting body states and processes such as glycaemia-induced oxidative stress and inflammation [4], which are implicated in the development of diabetic complications [5]. It is therefore important that the need to control intakes of glycaemic carbohydrate should not be misconstrued as a need to decrease intakes of fruits, particularly as fruit structure, fruit sugar (fructose), and other fruit constituents that retard glycaemic response, such as cell wall debris (dietary fibre), organic acids, and phenolics acting in the gut, may lead to a relatively low glycaemic response to sugars in fruit. We have shown that in kiwifruit, both the available carbohydrates and other components, contribute to an improved glycaemic response upon equal-carbohydrate partial substitution of kiwifruit for starchy breakfast cereal [6].

One of the strategies for glycaemia management that has been used in dietetic practice is to substitute highly glycaemic carbohydrates in the diet with less glycaemic carbohydrates using the “carbohydrate exchange” system [7]. Carbohydrate exchanges are generally carried out by substituting low glycaemic impact carbohydrate for higher glycaemic impact carbohydrate sources, while maintaining a more or less constant nutrient composition. However, rather than simply substituting one carbohydrate source for another, carbohydrate exchanges provide an opportunity to improve the nutrient richness of the diet by partially substituting nutrient-rich carbohydrate products, such as fresh kiwifruit, for less nutrient-rich foods such as starch-based staples and refined cereal products.

To facilitate the more general use of kiwifruit in food exchange for nutrient enrichment or glycaemic control we determined the available carbohydrate content of two cultivars of kiwifruit (*Actinidia chinensis* var. *deliciosa* “Hayward” marketed as Zespri^®^ Green kiwifruit, and *Actinidia chinensis* var. *chinensis* “Zesy002”, marketed as Zespri^®^ SunGold kiwifruit) by in vitro digestive analysis, and their glycaemic potencies relative to a glucose reference in a human intervention study. The results allowed quantities to be calculated for either equi-carbohydrate or equi-glycaemic substitution of a range of foods, and construction of tables of kiwifruit exchanges to show:
(1)How much of a range of carbohydrate foods could be substituted by one kiwifruit while keeping available carbohydrate content constant.(2)Approximately how much glycaemic impact would be altered by equal carbohydrate partial substitution of one kiwifruit for a food.(3)Approximately how much of a number of foods could be substituted by one kiwifruit while keeping glycaemic impact constant.


The aim of the study was to demonstrate the feasibility and glycaemic safety of using kiwifruit exchanges to increase intake of nutrients such as vitamin C within the diet with little effect on its glycaemic impact or carbohydrate load.

## 2. Methods

### 2.1. Test Components

“Hayward” (GR) and “Zesy002” (SG) kiwifruit were provided by Zespri Group Ltd., Tauranga, New Zealand, in a ready-to-eat state of ripeness, and processed within a few days of receipt. They were peeled and the hard apical core removed from the green kiwifruit, then halved and frozen (−20 °C). The frozen fruit were allowed to thaw partially and were then crushed to a coarse pulp by briefly (10 s) chopping in a Halde food processor. The pulp was then divided accurately into individual 200-g portions, each stored frozen within a plastic, capped, freezer-proof sundae container until required.

The glucose used was dextrose monohydrate (Davis Food Ingredients, Palmerston North, New Zealand), which contains 91% glucose. It is henceforth referred to as glucose, and an allowance was made for its water content in all calculations and weight measurements.

### 2.2. Carbohydrate Analysis of Kiwifruit

The available carbohydrate contents of the fruit (GR and SG) were determined by a standard digestive analysis, involving simulated gastric (pepsin pH 2.5, 30 min) and small intestinal digestion (pancreatin/amyloglucosidase, pH 6.2, 120 min), with the available carbohydrate content of the digested pulp measured after invertase digestion using a reduced scale modification of the dinitrosalicylic acid method [8]. The fructose content was measured by the thiobarbituric acid procedure [9].

### 2.3. Formulation of Meals

The subjects consumed three materials:
(1)40 g of glucose (reference).(2)200 g of “Hayward” kiwifruit alone(3)200 g of “Zesy002” kiwifruit alone.


### 2.4. Human Intervention Study

This study was conducted according to the guidelines laid down in the Declaration of Helsinki and all procedures involving human subjects were approved by the Human and Disabilities Ethics Committee of the New Zealand Ministry of Health (ethics approval number 14/CEN/207). Written informed consent was obtained from all subjects. The trial was registered with the Australia New Zealand Clinical Trials Registry (Trial ID: ACTRN12615000222549) (URL: http://www.anzctr.org.au). The participant flow chart shows ethical approval, recruitment, and intervention processes for the trial (Figure 1).

The trial was run as a non-blinded randomised repeated measures study. It was not possible to blind the subjects to the meals they were consuming. However, the data and statistical analysis were performed by an analyst who was blinded to the treatments. Meal order was randomised for each subject using a computerised random number generator.

Participants: Twenty participants, eight male and 12 female, were recruited by flyer and email. Respondents were interviewed and given an information pack including a description of the study and a consent form. Prospective participants were asked to complete a health questionnaire and provide a capillary blood glucose sample for glucose and glycated haemoglobin analysis. Exclusion criteria included known intolerance of kiwifruit, glucose intolerance as indicated by the fasting blood glucose and HbA1c, and recent ill health. The characteristics (mean ± SD) of the study group were as follows: age 36.7 ± 8.1 years, BMI 24.5 ± 5.2 kg.m^−2^, fasting glucose 4.6 ± 0.4 mmol.L^−1^, HbA1c 33.9 ± 4 mmol.mol^−1^).

Ingestion: The kiwifruit were thawed in a microwave with care to avoid heating, immediately before being consumed. The glucose and kiwifruit were consumed with enough water to give an equal intake volume (300 mL) in all meals.

Glycaemic response: Subjects were asked to eat moderately the evening before and to fast overnight and present themselves at 0830 h for the dietary intervention. They were asked to consume the meals within a 10-min period and avoid physical exertion for three hours afterward, during which time blood glucose determinations were made. Blood glucose concentrations were measured by finger-prick analysis of capillary blood using a HemoCue (Ängelholm, Sweden) blood glucose analyser. Blood samplings were made immediately before consuming the diets (duplicate, baseline), and at 15, 30, 45, 60, 90, 120, and 180 min after the start of food consumption.

Plasma insulin: Plasma insulin was determined with a Human Insulin Elisa kit (Millipore Hertfordshire, UK) using 0.8-mL capillary blood glucose samples collected at the same time as capillary blood glucose measurements of the glucose reference and SG kiwifruit. A subset of six subjects was used for insulin determination, with their response to SG compared with response to the glucose reference. The aim was to show that kiwifruit exchange did not induce a disproportionately high insulin response. An obvious outlier was removed from the glucose and kiwifruit groups because of technical problems with the glucose analysis, so the analysis involved five subjects. Removal of the outlier did not affect conclusions drawn from the results.

### 2.5. Data Analysis

Glycaemic potency of kiwifruit carbohydrate: Incremental blood glucose responses were calculated by subtracting each individual’s average baseline value from their subsequent measurements and were then used to determine the incremental area under the curve (IAUC) for each individual by trapezoid summation of positive increments [10]. Data were entered into a Microsoft^®^ Excel spreadsheet for preliminary analysis. For statistical comparison of means (ANOVA), GenStat software was used (version 11.1, VSNi Ltd., Hemel Hempstead, UK). Data were analysed using unbalanced analysis of variance (ANOVA), testing differences between meals after adjusting for participant and order effects.

Relative glycaemic potency of the carbohydrate (CHO) in kiwifruit (grams of glucose equivalents (GGEs)/100 g CHO) was used as an estimate of glycaemic index and calculated from the areas under the blood glucose response curves for the kiwifruit (IAUC_200 g KF_) relative to the glucose reference (IAUC_40 g Glucose_), with adjustment for weights of carbohydrate involved (Table 1):Relative glycaemic potency (RGP) = (IAUC_KF_/IAUC_glucose_) × (Wt. glucose/Wt. KF) × 100 GGE/100 g CHO(1)
RGP = (IAUC_200 g KF_/IAUC_40 g Glucose_) × (40/200) × 100  GGE/100 g CHO(2)

GGE expresses the effects of foods on blood glucose on a whole-food basis relative to glucose, as grams of glucose equivalents (GGEs) [11]. The GGE has weight (g) units, so it can be expressed per serving or per reference amount customarily consumed, or per 100 g of food. Thus, it behaves like a nutrient value, and has been termed a virtual food component [12]. It allows direct comparison of foods, and indicates how much glucose a food quantity would be equivalent to in its glycaemic effect. A food composition table containing GGE values allows one to see not only what a food is, but also what it does in terms of its relative glycaemic effect.

Kiwifruit equivalents and kiwifruit exchanges: From a knowledge of the available carbohydrate and GGE content of various foods, including kiwifruit, it was possible to construct tables as guides to incorporating kiwifruit into diets while managing postprandial glycaemia as follows. Because the glycaemic potencies of the two kiwifruit cultivars were similar, the exchange tables have been based on the SunGold values for carbohydrate (12.3 g/100 g) and GGE (6.7 g/100 g) content.

(1) Equi-carbohydrate exchanges (Table 2)

The amount of a food that could be exchanged for (substituted by) one kiwifruit without altering carbohydrate intake was calculated from the available carbohydrate content (%) of kiwifruit and the substituted food.

Where %CHO_f_ is the percentage of available carbohydrate in the food (New Zealand Food Composition Database), and %CHO_k_ is the percentage of carbohydrate in the kiwifruit, 100 g kiwifruit (one kiwifruit) would exchange: (%CHO_k_/%CHO_f_) × 100 g of the food, or ((%CHO_k_/%CHO_f_) × 100)/Sf servings of the food, where Sf is the serving size (g) of the food.

(2) Glycaemic impact of carbohydrate-based kiwifruit exchange

From the glycaemic potencies (GGE contents) of kiwifruit and foods, and their carbohydrate contents, one may estimate the reduction in relative glycaemic impact that could be expected by equi-carbohydrate partial substitution of kiwifruit for a food. This would allow estimation of possible changes in glycaemic response, for the purposes of blood glucose management.

The proportional reduction in glycaemic impact as a result of substitution by kiwifruit is the relative glycaemic potency of the food (GGE/serving) after substituting plus the GGE added in the substituting kiwifruit, as a proportion of the relative glycaemic potency of the unsubstituted food. The relative glycaemic potency of the food (RGP_f_) may be estimated as glycaemic load from its carbohydrate content and glycaemic index (Table 1).

The glycaemic potency of the food before substituting (GGE_B_) is:(GGE_B_) = Wt food × RGP_f_/100(3)

After substituting (GGE_A_), it is:GGE_A_ = ((Wt food × % CHO) − CHO_KF_) × RGP_f_/% CHO(4)
where CHO_KF_ is the carbohydrate in the substituting kiwifruit. Including the GGE contribution from the substituting kiwifruit (6.7 GGE), the proportional (%) decrease as a result of substitution would be:% decrease in GGE = (GGE_A_ + 6.7) × 100/GGE_B_(5)

(3) Equi-glycaemic partial substitution (Table 3)

From the relative glycaemic potencies of kiwifruit and a food (GGE/100 g = RGP) one can estimate the amount of the food that should be removed from a meal to include a kiwifruit without increasing postprandial glycaemia. This approach may be useful where the aim is to include fruits in the diet while maintaining a constant blood glucose response.

If the RGP of a food is RGP_f_, and the RGP of kiwifruit is RGP_KF_, the weight of food (W_f_) equivalent to 100 g of kiwifruit is given by:W_f_ = RGP_f_/RGP_KF_ × 100(6)

Similarly, the weight of kiwifruit (W_KF_) that is the glycaemic equivalent of any given food quantity (W_f_), such as a serving, is easily calculated from the relationship between the relative glycaemic potencies of the kiwifruit (RGP_KF_) and the food (RGP_f_):W_KF_ = W_f_ × RGP_KF_/RGP_f_(7)

This information may be useful where it is desired to replace a complete item with kiwifruit without altering glycaemic response.

## 3. Results

Analysis of kiwifruit: Digestive analysis showed the available carbohydrate contents of the kiwifruit to be as follows: “Hayward” green kiwifruit, 11.2% *w*/*w*; “Zesy002” gold kiwifruit, 12.3% *w*/*w*.

The figures were close to values from previous analyses of six cultivars of kiwifruit (New Zealand Food Composition Database). The sugars consisted of approximately equal proportions of glucose and fructose, with a lesser sucrose component, in the approximate ratio 2:2:1.

Blood glucose responses: All 20 subjects completed the trial and the results from all of them were used in the data analysis. The between-subject variations were large, as is typical of such studies, but no outliers were removed. The different diets induced blood glucose responses that were clearly distinctive (Figure 2). The responses to the two kiwifruit cultivars were very similar and less than for the glucose reference.

Relative glycaemic potency: The relative glycaemic potency (RGP; grams of glucose equivalents (GGE)) of the whole fruit, calculated from the glycaemic response to 200 g kiwifruit compared with the response to 40 g of glucose reference (Figure 2), and based on the area under the blood glucose response curve, showed that in terms of blood glucose-raising potential, one 100-g piece of “Hayward” kiwifruit would have a blood glucose-raising (glycaemic) potency equivalent to that of 6.6 g of glucose, and one 100-g piece of “Zesy002” kiwifruit would have a glycaemic potency equivalent to 6.7 g glucose (Table 4).

The RGP of the available carbohydrate alone in “Hayward”, its approximate glycaemic index, was 59 ± 7.03 GGE/100 g carbohydrate (mean ± SEM) and that of “Zesy002” was 54 ± 3.05 GGE/100 g carbohydrate (mean ± SEM), with the difference between the values non-significant.

Insulin responses: On an equal carbohydrate basis, insulin response was lower for kiwifruit carbohydrate than for than for glucose (Table 5), consistent with the lower insulinaemic potential of fructose compared with glucose [14]. A lower insulin response to kiwifruit than to rice in equal carbohydrate meals has similarly been measured recently [15]. When expressed per GGE the insulin responses were very similar (Table 5), indicating that it is physiologically valid to express glycaemic potency of mixed sugars in fruit as glucose equivalents, as indicated in previous studies of the relative effects of glucose and fructose on the insulin response [14].

With the one removed outlier subject included in the analysis the median insulin response (μU mL^−1^.min^−1^) per gram of carbohydrate was: glucose 39.7, SunGold 24.1; and the insulin response per GGE was: glucose 39.7, SunGold 42.1.

## 4. Discussion

The tables of equi-carbohydrate (Table 2) and equi-glycaemic (Table 3) exchanges of kiwifruit show that kiwifruit exchanges over a range of food categories will result in very small changes in glycaemic impact, based purely on the glycaemic potency of the sugars involved. The change in GGE intake using one kiwifruit exchange is within the range ± 5 GGE for most of the foods considered (Figure 3). Because the exchange is on a carbohydrate basis, whether kiwifruit exchange increases or decreases glycaemic impact will depend on the glycaemic index (GI) of the food being substituted relative to the GI of kiwifruit. Substitution of any foods with a GI less than that of kiwifruit (GI = 54) will increase glycaemic impact, and substitution of foods with a GI greater than 54 will reduce glycaemic impact. However, because the substitution involves quite small amounts of carbohydrate, due to the high water content of kiwifruit and the fact that kiwifruit has a low GI, the change in GGE intake will be small. Food groups in which kiwifruit substitution would slightly increase glycaemic impact include pasta, pulses, and some fruits. Kiwifruit substitution of cereal-based starchy foods such as bakery products and breakfast cereals would reduce glycaemic impact most.

The exchange tables indicate the change in relative glycaemic impact that may be attributed to carbohydrate exchange (Table 2, Figure 3). However, kiwifruit substitution is likely to cause a greater reduction in glycaemic impact than would be predicted from carbohydrate substitution alone because of the presence of organic acids, dietary fibre, and other fruit components such as phenolics. Thus, where kiwifruit exchange indicates an increase in relative glycaemic impact, as in the case of pulses and pasta, the increase is likely to be smaller than indicated in Figure 3. Furthermore, the pattern of food intake can have a sizeable effect on glycaemic response: if the kiwifruit portion of the substituted meal is consumed 30 min before the rest of the meal, the overall glycaemic response can be substantially suppressed [15], by as much as 30%.

The results of insulin analysis in a subset of participants (Table 5) showed that the insulin response per kiwifruit GGE was almost identical to the response per gram of glucose. Therefore, substitution of kiwifruit carbohydrate for food carbohydrate would not lead to a disproportionately large increase in insulin response compared with glucose. Because the exchange involves no more than 6–7 GGEs per kiwifruit, any disproportionate increase in insulin per GGE from kiwifruit would in any case give a small net change in insulin for the meal. Kiwifruit exchanges will not, therefore, appreciably increase insulin demand.

The vitamin C content of the kiwifruit was not determined for the present study because it has been previously well established as consistently higher than in most other fruits [1], with a concentration of about 150 mg per 100 g SunGold fruit—enough for a single fruit to raise vitamin C intake to the recommended daily allowance. The present analysis of the effects of kiwifruit exchanges has shown that using kiwifruit to improve intakes of vitamin C or other fruit components would have very little effect on glycaemic impact over a range of food categories, and is therefore glycaemically safe. With the aid of exchange tables, kiwifruit may be used to improve vitamin C intakes while maintaining either a constant carbohydrate intake, or a more or less constant glycaemic impact and insulin demand.

Although the focus has been on vitamin C in this paper, it is noteworthy that kiwifruit have a high content of potassium, 315 mg per 100 g (about one fruit) [1]. When kiwifruit exchange involves substitution of refined cereal products, including many breakfast cereals, a substantial increase in potassium may result. For instance, one cup of cornflakes containing 24.4 g of available carbohydrate provides 27 mg of potassium. Exchange for two SunGold kiwifruit providing 24.6 g of available carbohydrate, according to the analysis in this paper, would provide 630 mg of potassium, a 24-fold increase. Similarly, substitution of highly refined cereal products by kiwifruit on an equal carbohydrate basis may help to address the shortfall in dietary fibre in many modern diets.

The utility of the exchange tables in dietary management of glycaemia, and how accurately they achieve this purpose, requires further validation, particularly for the equi-glycaemic exchanges which are based on glycaemic response. The equal carbohydrate exchanges can be conducted accurately, because the measurements on which they are based are purely chemical analyses of available carbohydrate content of foods. If measured directly, available carbohydrate is accurate, although it is often measured indirectly “by difference”, which is not as accurate. However, measures involving glycaemic responses such as GGEs, RGP, and GI involve intrinsically larger errors [16], arising from individual differences and variations in physiological response to foods.

Calculating the glucose equivalence of 200 g kiwifruit with a 40-g glucose reference involves a small error because of the non-linearity of the glucose dose–blood glucose response curve [11]. The present study was about the glycaemic impact of foods, not about GI per se, so there was no need to use a 50-g glucose reference as specified for GI determination. Instead, the reference glucose dose of 40 g was closer to the carbohydrate dose of 23 g ingested in the realistic 200 g intake of kiwifruit, which is the edible portion of two fresh kiwifruit, used in the clinical trial of the present study. Therefore, no adjustment for non-linearity was required when calculating the glucose equivalence (GGE content) of the kiwifruit [17]. We have also shown that the difference between GGE estimated as glycaemic load from GI and carbohydrate content, and GGE measured directly, is small [18]. Thus, glycaemic load values may be used as estimates of RGP with little loss of accuracy in guiding food choices for glycaemia management [19].

The present study has shown the glycaemic potencies of “Hayward” and “Zesy002” in comparison to a glucose reference (their relative glycaemic potencies) to be very low. Both kiwifruit cultivars had an RGP of 6–7 GGEs per 100 g. That is, 100 g (one kiwifruit edible portion) would have the same glycaemic effect as 6–7 g of glucose. Converting the GGE content to a per 100-g carbohydrate basis gave estimates of GI for “Hayward” and “Zesy002” of 59 and 54, respectively. The values were slightly higher than previously published [20] perhaps because the fruit were consumed after disintegration and freezing. However, with the very low relative glycaemic potency and low available carbohydrate content of kiwifruit, a difference of 10 GI units would translate to a difference of about 1 g of glucose equivalents per kiwifruit. With a carbohydrate content of about 10%, a difference of 10 GI units would make a difference of only about 1 GGEs (the effect equivalent to 1 g glucose) in a 100 g edible portion of kiwifruit. This underlines the irrelevance of GI to management of intakes of kiwifruit and other fresh fruits for postprandial glycaemic control, and the need to use more realistic means of expressing glycaemic potency, such as RGP. So even if consumed on an equal carbohydrate basis as a partial replacement of low GI foods there would be little glycaemic cost compared with the benefit of a greatly increased intake of vitamin C.

Similarly, when kiwifruit is used in an exchange format to partially substitute other carbohydrate foods, the net effect on GGE intake of removing carbohydrate in the substituted food and adding it in the substituting kiwifruit is only a few GGE units (Figure 3). Furthermore, partially substituting fresh kiwifruit, with a vitamin C content of about 100 mg per fruit, for starch-based foods such as breakfast cereals and cereal-based cooked staples, has the potential to enormously improve vitamin C status. As vitamin C is heat-labile and most cereal products and other starchy staples are cooked and therefore contain very low amounts of vitamin C, partial substitution by one or two kiwifruit would be a useful strategy for naturally improving vitamin C intakes.

Substituting kiwifruit for unsweetened starchy products would increase fructose intakes slightly, but by an amount that would not produce the metabolic changes that are associated with high fructose intakes in sweetened processed foods [21]. In fact, a modest intake of fructose has the benefit of enhancing glucose metabolism and facilitating glucose disposal [22]. We have found that consuming two kiwifruit per day for 12 weeks did not cause any of the metabolic changes that have been reported for high intakes of fructose (paper in preparation).

The present paper has illustrated the concept of kiwifruit exchange tables, but the practicality of using the exchange tables, and the range of foods they contain also needs further development. While the tables in their present form would be easily understood by nutritionists and dieticians, how the information could be best used in a user-friendly format for public use, could be the subject of consumer research. Further research in long-term trials should also be conducted to determine whether or not sustained use of kiwifruit exchanges leads to improvements, or delays decline, in key biomarkers of health outcomes.

The exchange tables are based on a value for ready-to-eat ripe fruit, but such a value is likely to depend on the stage of ripeness of the fruit when consumed, so it is also important to determine the stability of the GGE values determined in the present study. In that respect, the values are likely to be similar to any other values in a food composition database that are guides, but do not pretend to be exact predictors of effects of any given food ingestion event.

Overall, the results indicate that including kiwifruit or other fruits in diets by equi-carbohydrate substitution of highly digestible and therefore high GI starch components will generally lead to glycaemic benefits, while also enriching the diets functionally and nutritionally. At the very least, despite the perception of fruits as sweet-flavoured and therefore high in glycaemic sugars, the glycaemic change associated with consuming fresh fruits in a carbohydrate exchange format is small. Including fresh fruit in the diet need not have a negative glycaemic impact when its introduction is guided by tables of exchanges based on carbohydrate or glycaemic equivalents, as presented here for kiwifruit.

## Figures and Tables

**Figure 1 nutrients-10-01710-f001:**
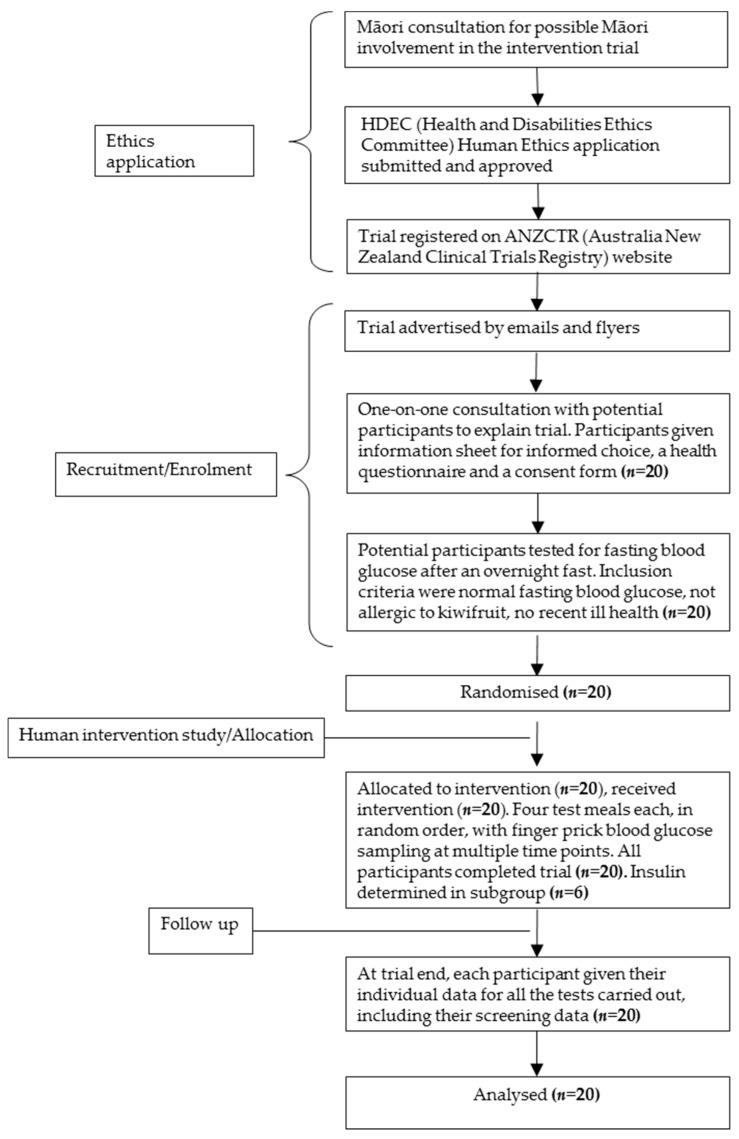
Participant flow chart for clinical determination of the relative glycaemic potency of kiwifruit.

**Figure 2 nutrients-10-01710-f002:**
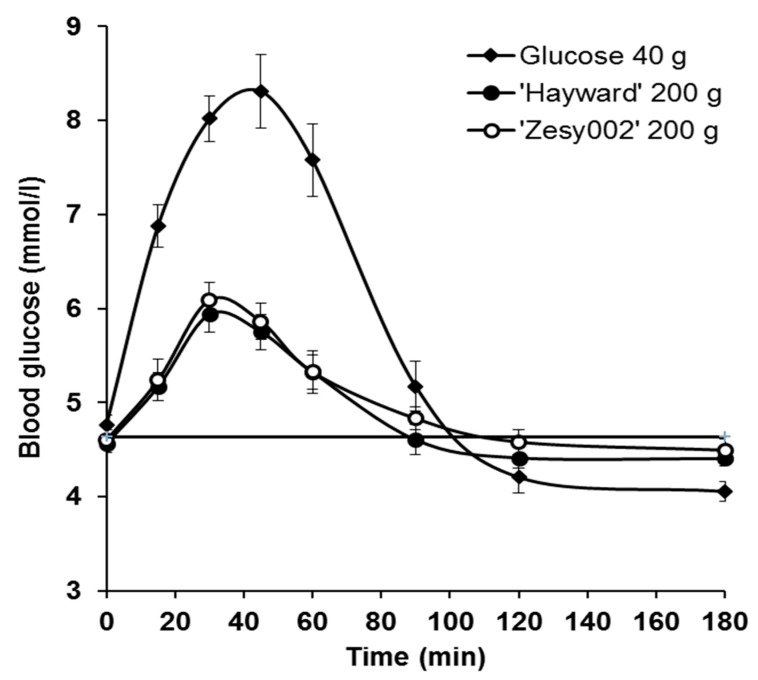
Mean (± SEM) blood glucose responses used to determine the relative glycaemic potency of “Hayward” (GR) and “Zesy002” (SunGold, SG) kiwifruit.

**Figure 3 nutrients-10-01710-f003:**
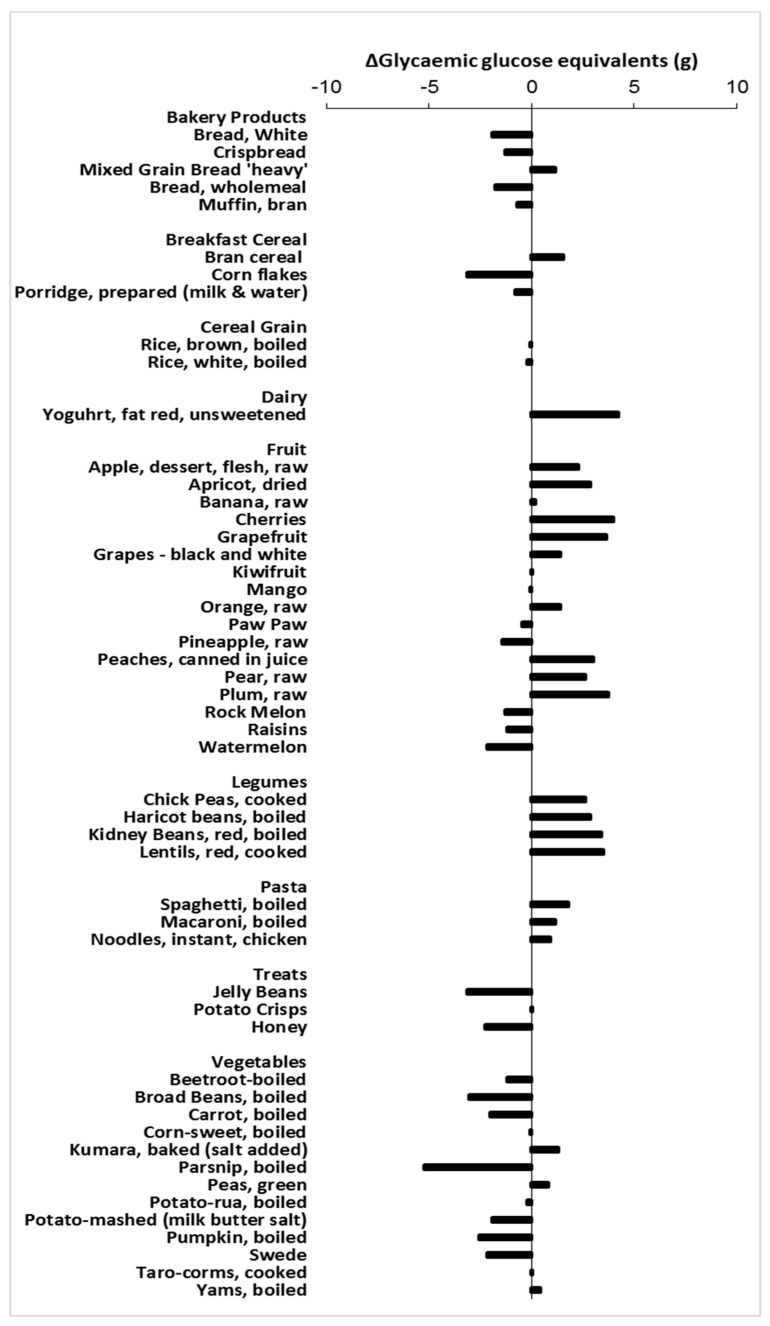
Change in glycaemic potential (GGE (g)) upon equi-carbohydrate substitution of one SunGold kiwifruit for various foods.

**Table 1 nutrients-10-01710-t001:** Estimation of relative glycaemic potency (RGP) of foods as glycaemic load for a selection of foods with available glycaemic index (GI) and available carbohydrate (% CHO avail.) values.

Description	Portion Size ^1^	Weight ^2^	GI ^3^	CHO Avail.	RGP ^4^	GGEs/CSM ^5^
	(CSM)	(g)	(%)	(%)	(GGE/100 g)	
**Bakery Products**						
Bread, white	1 medium slice	26	70	43.4	30.4	7.9
Crispbread	1 biscuit	6	65	64.4	41.9	2.5
Mixed grain bread “heavy”	1 medium slice	28	45	36.7	16.5	4.6
Bread, wholemeal	1 medium slice	28	69	37.1	25.6	7.2
Muffin, bran	1 muffin	80	60	32.5	19.5	15.6
**Breakfast Cereal**						
Bran cereal	1 cup	45	42	37.4	15.7	7.1
Corn flakes	1 cup	31	80	84.8	67.8	21.0
Porridge, prepared	1 cup	260	61	10.5	6.4	16.7
**Cereal Grain**						
Rice, brown, boiled	1 cup	206	55	29.2	16.1	33.1
Rice, white, boiled	1 cup	144	56	17.5	9.8	14.1
**Dairy**						
Yoghurt, fat red, unsweetened	1 pottle	150	20	14.8	3.0	4.4
**Fruit**						
Apple, dessert, flesh, raw	1 apple	130	36	10.7	3.9	5.0
Apricot, dried	10 halves	35	31	48.8	15.1	5.3
Banana, raw	1 banana	128	53	24.1	12.8	16.3
Cherries	10 cherries	67	22	14.0	3.1	2.1
Grapefruit	1 grapefruit	236	25	10.1	2.5	6.0
Grapes, black and white	10 grapes	54	43	15.8	6.8	3.7
Kiwifruit	1 kiwi fruit	100	52	10.0	5.2	5.2
Mango	1 cup slices	176	55	14.6	8.0	14.1
Orange, raw	1 orange	128	43	7.7	3.3	4.2
Pawpaw	1 slice	140	58	6.9	4.0	5.6
Pineapple, raw	1 cup chopped	164	66	11.4	7.5	12.3
Peaches, canned in juice	1 cup slices	264	30	22.2	6.7	17.6
Pear, raw	1 pear	148	33	12.8	4.2	6.3
Plum, raw	1 plum	49	24	13.9	3.3	1.6
Rock melon	1 cup sliced	168	65	6.4	4.1	7.0
Raisins	1 cup	154	64	71.3	45.6	70.3
Watermelon	1 cup flesh	169	72	5.1	3.7	6.2
**Legumes**						
Chickpeas, cooked	1 cup	173	33	8.6	2.8	4.9
Haricot beans, boiled	1 cup	180	31	15.2	4.7	8.5
Kidney beans, red, boiled	1 cup	187	27	15.9	4.3	8.0
Lentils, red, cooked	1 cup	209	26	10.4	2.7	5.7
**Pasta**						
Spaghetti, boiled	1 cup	148	40	23.9	9.6	14.1
Macaroni, boiled	1 cup	149	45	16.8	7.6	11.3
Noodles, instant, chicken	1 packet	200	47	5.8	2.7	6.0
**Treats**						
Jelly beans	10 jelly beans	20	80	91.8	73.4	14.7
Potato crisps	1 small packet	50	54	47.6	25.7	12.9
Honey	1 tablespoon	21	73	79.6	58.1	12.2
**Vegetables**						
Beetroot, boiled	1 cup slices	180	64	9.8	6.3	11.3
Broad beans, boiled	1 cup	170	79	8.6	6.8	11.5
Carrot, boiled	1 carrot	49	71	5.5	3.9	1.9
Corn-sweet, boiled	1 cob	128	55	20.9	11.5	14.7
Kumara, baked	1 kumara	114	44	23.3	10.3	11.7
Parsnip, boiled	1 parsnip	160	97	12.3	11.9	19.1
Peas, green	1 cup	164	48	7.1	3.4	5.6
Potato, rua, boiled	1 potato	114	56	18.2	10.2	11.6
Potato, mashed	1 cup	209	70	14.5	10.2	21.2
Pumpkin, boiled	1 cup	220	75	4.0	3.0	6.6
Swede	1 cup chopped	150	72	3.7	2.7	4.0
Taro corms, cooked	1 cup sliced	142	54	27.4	14.8	21.0
Yams, boiled	1 cup cubes	144	51	27.1	13.8	19.9

^1^ Portion size used was a “Common Standard Measure” (CSM); ^2^ Weight of a portion (g); ^3^ From the international tables of glycaemic index and glycaemic load a [13]; ^4^ GI∙% available carbohydrate (CHO avail (%)); ^5^ Relative glycaemic impact per portion expressed as glucose equivalents.

**Table 2 nutrients-10-01710-t002:** Kiwifruit exchanges that would maintain a constant carbohydrate intake, based on the available carbohydrate content of selected carbohydrate foods.

Description	Portion Size	Weight	SG Equivalents		1 Zespri^®^ SunGold Kiwifruit (SG) Replaces		GGE
		(g)	per 100 g ^1^	per Portion ^2^	(g) ^3^	Portions ^4^	(g) ^5^
**Bakery Products**							
Bread, white	1 medium slice	26	3.53	0.9	28	1.09	−1.9
Crispbread	1 biscuit	6	5.24	0.3	19	3.18	−1.3
Mixed grain bread	1 medium slice	28	2.98	0.8	34	1.20	1.2
Bread, wholemeal	1 medium slice	28	3.02	0.8	33	1.18	−1.8
Muffin, bran	1 muffin	80	2.64	2.1	38	0.47	−0.7
**Breakfast Cereal**							
Bran cereal	1 cup	45	3.04	1.4	33	0.73	1.5
Corn flakes	1 cup	31	6.89	2.1	15	0.47	−3.1
Porridge, prepared	1 cup	260	0.85	2.2	117	0.45	−0.8
**Cereal Grain**							
Rice, brown, boiled	1 cup	206	2.37	4.9	42	0.20	−0.1
Rice, white, boiled	1 cup	144	1.42	2.0	70	0.49	−0.2
**Dairy**							
Yoghurt, fat red, unsweetened	1 pottle	150	1.20	1.8	83	0.55	4.2
**Fruit**							
Apple, dessert, raw	1 apple	130	0.87	1.1	115	0.88	2.3
Apricot, dried	10 halves	35	3.97	1.4	25	0.72	2.9
Banana, raw	1 banana	128	1.96	2.5	51	0.40	0.2
Cherries	10 cherries	67	1.14	0.8	88	1.31	4.0
Grapefruit	1 grapefruit	236	0.82	1.9	122	0.52	3.6
Grapes	10 grapes	54	1.28	0.7	78	1.44	1.4
Kiwifruit	1 kiwi fruit	100	1.00	1.0	100	1.00	0
Mango	1 cup slices	176	1.19	2.1	84	0.48	−0.1
Orange, raw	1 orange	128	0.63	0.8	160	1.25	1.4
Pawpaw	1 slice	140	0.56	0.8	178	1.27	−0.4
Pineapple, raw	1 cup	164	0.93	1.5	108	0.66	−1.4
Peaches, canned	1 cup slices	264	1.80	4.8	55	0.21	3.0
Pear, raw	1 pear	148	1.04	1.5	96	0.65	2.6
Plum, raw	1 plum	49	1.13	0.6	88	1.81	3.7
Rock Melon	1 cup sliced	168	0.52	0.9	193	1.15	−1.3
Raisins	1 cup	154	5.80	8.9	17	0.11	−1.2
Watermelon	1 cup flesh	169	0.41	0.7	241	1.43	−2.2
**Legumes**							
Chickpeas, cooked	1 cup	173	0.70	1.2	143	0.83	2.6
Haricot beans, boiled	1 cup	180	1.24	2.2	81	0.45	2.9
Kidney beans, boiled	1 cup	187	1.29	2.4	77	0.41	3.4
Lentils, red, cooked	1 cup	209	0.85	1.8	118	0.57	3.5
**Pasta**							
Spaghetti, boiled	1 cup	148	1.94	2.9	51	0.35	1.8
Macaroni, boiled	1 cup	149	1.37	2.0	73	0.49	1.2
Noodles, instant	1 packet	200	0.47	0.9	212	1.06	0.9
**Treats**							
Jelly beans	10 jelly beans	20	7.46	1.5	13	0.67	−3.1
Potato crisps	1 small packet	50	3.87	1.9	26	0.52	0.1
Honey	1 tablespoon	21	6.47	1.4	15	0.74	−2.3
**Vegetables**							
Beetroot, boiled	1 cup slices	180	0.80	1.4	126	0.70	−1.2
Broad beans, boiled	1 cup	170	0.70	1.2	143	0.84	−3.0
Carrot, boiled	1 carrot	49	0.45	0.2	224	4.56	−2.0
Corn, sweet, boiled	1 cob	128	1.70	2.2	59	0.46	−0.1
Kumara, baked	1 kumara	114	1.89	2.2	53	0.46	1.3
Parsnip, boiled	1 parsnip	160	1.00	1.6	100	0.63	−5.2
Peas, green	1 cup	164	0.58	0.9	173	1.06	0.8
Potato, rua, boiled	1 potato	114	1.48	1.7	68	0.59	−0.2
Potato, mashed	1 cup	209	1.18	2.5	85	0.41	−1.9
Pumpkin, boiled	1 cup	220	0.33	0.7	308	1.40	−2.5
Swede	1 cup	150	0.30	0.5	332	2.22	−2.2
Taro corms, cooked	1 cup sliced	142	2.23	3.2	45	0.32	0.1
Yams, boiled	1 cup cubes	144	2.20	3.2	45	0.32	0.4

^1^ % carbohydrate in food/12.3 (12.3 = % available carbohydrate in kiwifruit); ^2^ (Portion weight × % carbohydrate in food/100)/12.3; ^3^ (12.3 × 100)/% available carbohydrate in food; ^4^ ((12.3 × 100)/% available carbohydrate in food)/portion weight; ^5^ 6.7—(12.3 × GI of food/100). GGEs = grams of glucose equivalents.

**Table 3 nutrients-10-01710-t003:** Equi-glycemic kiwifruit exchanges based on the relative glycaemic potency of Zespri^®^ SunGold Kiwifruit (SG) and a selection of carbohydrate foods for which relative glycaemic potency (RGP) could be estimated from GI and available carbohydrate as grams of glucose equivalents (GGEs)/100 g, to maintain a constant glycaemic response.

Description	Portion Size	Weight	SG Equivalents		1 SG Replaces	
		(g)	(per 100 g) ^1^	(per Portion) ^2^	(g) ^3^	(Portions) ^4^
**Bakery Products**						
Bread, white	1 medium slice	26	4.5	1.2	22	0.8
Crispbread	1 biscuit	6	6.2	0.4	16	2.7
Mixed grain bread “heavy”	1 medium slice	28	2.5	0.7	41	1.4
Bread, wholemeal	1 medium slice	28	3.8	1.1	26	0.9
Muffin, bran	1 muffin	80	2.9	2.3	34	0.4
**Breakfast Cereal**						
Bran cereal	1 cup	45	2.3	1.1	43	0.9
Corn flakes	1 cup	31	10.1	3.1	10	0.3
Porridge, prepared	1 cup	260	1.0	2.5	105	0.4
**Cereal Grain**						
Rice, brown, boiled	1 cup	206	2.4	4.9	42	0.2
Rice, white, boiled	1 cup	144	1.5	2.1	68	0.5
**Dairy**						
Yoghurt, fat red, unsweetened	1 pottle	150	0.4	0.7	226	1.5
**Fruit**						
Apple, dessert, flesh, raw	1 apple	130	0.6	0.7	174	1.3
Apricot, dried	10 halves	35	2.3	0.8	44	1.3
Banana, raw	1 banana	128	1.9	2.4	52	0.4
Cherries	10 cherries	67	0.5	0.3	218	3.2
Grapefruit	1 grapefruit	236	0.4	0.9	265	1.1
Grapes, black and white	10 grapes	54	1.0	0.5	99	1.8
Kiwifruit	1 kiwifruit	100	1.0	1.0	97	1.0
Mango	1 cup slices	176	1.2	2.1	83	0.5
Orange, raw	1 orange	128	0.5	0.6	202	1.6
Pawpaw	1 slice	140	0.6	0.8	167	1.2
Pineapple, raw	1 cup chopped	164	1.1	1.8	89	0.5
Peaches, canned in juice	1 cup slices	264	1.0	2.6	101	0.4
Pear, raw	1 pear	148	0.6	0.9	159	1.1
Plum, raw	1 plum	49	0.5	0.2	201	4.1
Rock melon	1 cup sliced	168	0.6	1.0	162	1.0
Raisins	1 cup	154	6.8	10.5	15	0.1
Watermelon	1 cup flesh	169	0.5	0.9	182	1.1
**Legumes**						
Chickpeas, cooked	1 cup	173	0.4	0.7	236	1.4
Haricot beans, boiled	1 cup	180	0.7	1.3	142	0.8
Kidney beans, red, boiled	1 cup	187	0.6	1.2	156	0.8
Lentils, red, cooked	1 cup	209	0.4	0.8	248	1.2
**Pasta**						
Spaghetti, boiled	1 cup	148	1.4	2.1	70	0.5
Macaroni, boiled	1 cup	149	1.1	1.7	89	0.6
Noodles, instant, chicken	1 packet	200	0.4	0.9	246	1.1
**Treats**						
Jelly beans	10 jelly beans	20	11.0	2.2	9	0.5
Potato crisps	1 small packet	50	3.8	1.9	26	0.5
Honey	1 tablespoon	21	8.7	1.8	12	0.5
**Vegetables**						
Beetroot, boiled	1 cup slices	180	0.9	1.7	107	0.6
Broad beans, boiled	1 cup	170	1.0	1.7	99	0.6
Carrot, boiled	1 carrot	49	0.6	0.3	172	3.5
Corn, sweet, boiled	1 cob	128	1.7	2.2	58	0.5
Kumara, baked	1 kumara	114	1.5	1.7	65	0.6
Parsnip, boiled	1 parsnip	160	1.8	2.8	56	0.4
Peas, green	1 cup	164	0.5	0.8	197	1.2
Potato, rua, boiled	1 potato	114	1.5	1.7	66	0.6
Potato, mashed	1 cup	209	1.5	3.2	66	0.3
Pumpkin, boiled	1 cup	220	0.4	1.0	223	1.0
Swede	1 cup chopped	150	0.4	0.6	252	1.7
Taro corms, cooked	1 cup sliced	142	2.2	3.1	45	0.3
Yams, boiled	1 cup cubes	144	2.1	3.0	48	0.3

^1^ RGP (Table 1)/6.7; ^2^ GGE per CSM (Table 1)/6.7; ^3^ 6.7 × 100/RGP; ^4^ 6.7/(GGE per CSM (Table 1)).

**Table 4 nutrients-10-01710-t004:** Relative glycaemic potency (RGP) expressed as grams of glucose equivalents (GGEs) per 100 g of kiwifruit, determined from the relative areas under the blood glucose response curves (incremental area under the curve, IAUC) shown in Figure 2.

	IAUC		GGE/200 g		RRGP (GGEs/100 g) (GGEs/100 g)	
	Mean	SEM	Mean	SEM	Mean	SEM
Glucose (40 g)	234.7	18.8				-
“Hayward” kiwifruit (200 g)	75.1	7.3	13.2	1.25	6.6	0.8
“Zesy002” kiwifruit (200 g)	76.2	8.5	13.4	1.44	6.7	0.4

**Table 5 nutrients-10-01710-t005:** Insulin response to glucose and to glycaemic glucose equivalents (GGEs) in kiwifruit.

	Insulin Response (μU mL^−1^.min^−1^) per g Carbohydrate		Insulin Response (μU mL^−1^.min^−1^) per GGE	
	Mean	SEM	Mean	SEM
Glucose	40.9	2.2	40.9	2.2
Kiwifruit	28.8	5.3	39.6	4.5

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
