# Peer review of "Kiwifruit Exchanges for Increased Nutrient Richness with Little Effect on Carbohydrate Intake, Glycaemic Impact, or Insulin Response"

_nutrients, 2018, doi:10.3390/nu10111710_

Round 1
Reviewer 1 Report
The current study reported the equi-carbohydrate partial exchange of kiwifruit for starchy foods can greatly increase nutrient richness in a diet without increased glycemic and insulin response or carb intake. The methods and results were clearly and adequately presented. The conclusions from this study can be potentially meaningful for kiwifruit exchanges for other glycemic carbohydrate foods. Although there are certain practical limitations of the obtained data as all data were calculated purely from the carbohydrate substitution, without considering the effect of other food compositions on glycemic response. However, the authors fairly addressed this issue and some other implications of the study. Overall, this study is of merit for using kiwifruit exchange for nutrient richness and dietary management of glycaemia.
A few questions related with study design:
Why the relative glycemic potency of the carbohydrate in kiwifruit was calculated based on the glycemic response of 200 g kiwifruit compared with 40 g of glucose as reference?
Why glycemic response, AUC were measured during the time course of 3 hrs. rather than 2 hrs.?
Author Response
Thank you for your comments.
A 40 g reference was used instead of the 50 g usually used in glycaemic index determination because it was closer to the kiwifruit carbohydrate content, and is the reference amount that generally gives the most accurate relative glycaemic response values when the non-linearity of the glucose intake - blood glucose response is taken into account. This paper was not about GI or trying to provide GI values so there was no need to use a 50 g glucose reference. We discuss the reference at some length in the paper.
In our experience 120 min limit can lead to truncation of the glycaemic response curve, so 180 min is our preference in glycaemic response studies. We were not trying to determine GI so there was no need to adhere to 120 min. In any case, the responses had reached baseline at 120 min (Figure 2) so any time beyond that was not relevant.
Reviewer 2 Report
This manuscript reports on the assessment of the equi-carbohydrate and equi-glycemic partial exchanges of high glycemic foods with kiwifruit. In vitro digestion analysis was used to determine the available carbohydrate content, including the types of sugars and their quantities, of two types of kiwifruit. Glycemic potency was determined for the two types of kiwifruit, and a glucose reference, in a clinical trial that included 20 normoglycemic subjects. Based on these data and the available carbohydrate data from a governmental food composition database, exchange tables were developed, which provided theoretical equi-carbohydrate and equi-glycemic partial exchange calculations for replacing specific foods with kiwifruit. Based on the results of those calculations, the authors conclude that equi-carbohydrate partial exchange of kiwifruit, for selected starchy foods, can increase vitamin C intake while maintaining or lowering daily glycemic load.
This manuscript addresses topics of interest, including diet quality and glycemic control. In many socioeconomically developed nations, fruit intake is low. While select fruit products, such as fruit juices or processed fruits (e.g., canned fruits packed in heavy syrup), may have high carbohydrate and sugar contents, fresh fruits and unsweetened frozen fruits are generally nutrient-dense foods that provide many essential nutrients, antioxidants and fiber, for relatively few calories and modest quantities of sugar. Thus, assessment of the glycemic impact of kiwifruit with calculation of the expected impact of substituting kiwifruit for starchy foods has practical and scientific value.
This manuscript is well written overall, although clarification in a few sections would strengthen the interpretation of the results. Details of the areas needing clarification are discussed in the Specific Comments to the Authors section.
Specific Comments to the Authors
Methods—It would be useful to provide the conversion information for HbA1C from mmol/mol to % for readers who typically use % values.
Methods—the plasma insulin concentrations were calculated for 6 subjects as noted in the flow chart as well as stated in the text. The authors state that “An obvious outlier was removed from the glucose and kiwifruits groups, so the analysis involved six subjects.“ Additional information is needed as to why this subject was removed from the analyses. Were the results unphysiologic and therefore believed to be in error? How would the findings have been different had that subject been left in the analysis with reporting medians rather than means?
Results—the tables depict the results of the various calculations the authors performed. However, Table 1 - 3 are discussed in the methods section. The first mention of Table 4 is in the Discussion section. It seems that since these tables are results of the calculations, they need to be first introduced in the Results section.
Discussion—in more than one section, the authors mention the high vitamin C content of kiwifruit and provide it as an example of how kiwifruit intake can increase nutrient intake. And, as the authors state, intake of a kiwifruit provides the daily recommended amount of vitamin C. However, kiwifruits are a source of other nutrients too, such as fiber and potassium. Thus, the authors may want to consider providing a few examples of the change in fiber and/or potassium intake levels, alongside the glycemic impact, that results from the exchange of select foods with kiwifruit.
Author Response
Thank you for your comments.
With respect to the outlier, we have given the reason for removing them, in the methods section:
An obvious outlier was removed from the glucose and kiwifruit groups because of technical problems with the glucose analysis, so the analysis involved five subjects. Removal of the outlier did not affect conclusions drawn from the results.
In the results section we added the median results (average of the two middle results) with the outlier included. The pattern is similar to the results in Table 4.
With the one removed outlier subject included in the analysis the median results were: insulin response (μU mL-1.min-1) per gram of carbohydrate: Glucose 39.7, SunGold 24.1; and insulin response per GGE: Glucose 39.7, SunGold 42.1.
We would prefer to leave the first mention of Tables 1-3 in the methods section because we think it is useful to be able to point the reader to the data for which we are discussing calculations involved, in the methods section.
The first mention of table 4 is in fact in the results section. We have highlighted it.
We have added a paragraph to the discussion pointing out that kiwifruit exchange may improve intakes of other nutrients as well as Vitamin C:
Although the focus has been on Vitamin C in this paper, it is noteworthy that kiwifruit have a high content of potassium, 315 mg per 100 g (about one fruit) ([1]. When kiwifruit exchange involves substitution of refined cereal products, including many breakfast cereals, a substantial increase in potassium may result. For instance, one cup of cornflakes containing 24.4 g of available carbohydrate provides 27 mg of potassium. Exchange for two SunGold kiwifruit providing 24.6 g of available carbohydrate, according to the analysis in this paper, would provide 630 mg of potassium, a 24 fold increase. Similarly substitution of highly refined cereal products by kiwifruit on an equal carbohydrate basis may help to address the shortfall in dietary fibre in many modern diets.